

# Isolation and characterization of 10 polymorphic microsatellite loci for the endangered Galapagos-endemic whitespotted sandbass (*Paralabrax albomaculatus)*

Alicia C. Bertolotti[1], Sarah M. Griffiths[1], Nathan K. Truelove[2], Stephen J. Box[2], Richard F. Preziosi[1] and Pelayo Salinas de Leon[3]

[1] Faculty of Life Sciences, The University of Manchester, Manchester, United Kingdom
[2] Smithsonian Marine Station, Smithsonian, Fort Pierce Florida, United States
[3] Charles Darwin Research Station, Santa Cruz, Galapagos Islands, Ecuador

## ABSTRACT

The white-spotted sandbass (*Paralabrax albomaculatus)* is a commercially important species in the Galapagos Marine Reserve, but is classified as endangered in the IUCN Red List. For this study, 10 microsatellite loci were isolated and characterized using Illumina paired-end sequencing. These loci can be used for genetic studies of population structure and connectivity to aid in the management of the white-spotted sandbass and other closely-related species. The 10 characterized loci were polymorphic, with 11–49 alleles per locus, and observed heterozygosity ranged from 0.575 to 0.964. This set of markers is the first to be developed for this species.

## INTRODUCTION

The white-spotted sandbass (*Paralabrax albomaculatus* (Jenyns, 1840)) is a bony fish endemic to the Galápagos Islands. They have a pelagic larval stage that has the potential of wide dispersal throughout the archipelago. The white-spotted sandbass was listed as endangered in the IUCN Red List in 2001 following an estimated 70% decline in its population size, due mostly to overfishing (*Robertson et al., 2010*). Despite its endemism and importance for the Galápagos artisanal fishing community, very little research has focused on this species.

Fishing pressure is increasing for the white-spotted sandbass due to a rising human population (including an increase in tourist numbers; *Galapagos National Park, 2010*) and declines of previously favored species such as the sail-fin grouper (*Mycteroperca olfax*; *Ruttenberg, 2001*). Despite this decline, management regulations are severely limited, mostly by the lack of information about this endemic species. Species-specific genetic tools would vastly improve our ability to define population structure. In this study, we

Corresponding author
Alicia C. Bertolotti,
alicia.bertolotti@gmail.com

present 10 polymorphic microsatellites markers that have been developed specifically for this species. These primers can be used to determine genetic diversity, population structure and connectivity among populations in the archipelago.

## MATERIALS AND METHODS

All samples used in this study were collected with permission of the Galapagos National Park (research permit number: PC-28-13). Muscle or liver samples were collected from fishing boats at two distinct geographical locations: "Banco Ruso" ($n = 30$), south of San Cristobal island and "Bolivar" ($n = 40$) west of Isabela island. These fishing sites are approximately 220 km apart and are separated by the landmass of Isabela.

DNA was extracted from liver or muscle tissue using the DNEasy Blood and Tissue Kit (Qiagen, Venlo, Netherlands). To obtain microsatellite markers, a next-generation-sequencing approach was used (*Castoe et al., 2012a*). A paired-end library of genomic DNA was made with the Nextera® DNA Preparation Kit using 50 ng of DNA (following the manufacturers protocol) and sequenced on the Illumina MiSeq. The library construction and sequencing was carried out by the Genetics Core Facility at the University of Manchester. Read lengths were $2 \times 250$bp and there were $2 \times 4,238,835$ sequence reads obtained in the raw data. Microsatellites and their primers were then designed from reads filtered by Trimmomatic. Reads were trimmed using the 'sliding window' function based on quality scores with a 4bp window size and quality threshold of 20. Leading and trailing were both set to 3 and the minimum length was set to 50bp (see *Lohse et al., 2012*). There were $2 \times 3,930,136$ reads remaining after these filtering steps. Microsatellites with sufficient flanking region were screened for using PAL_finder v.0.02 (*Castoe et al., 2012b*). The primer settings were selected using the recommended criteria in the Qiagen Type-it Microsatellite PCR Kit protocol in order to increase amplification success for the development of primers when using this kit. These settings include: optimum base-pair length (bp) of 20–30pb, 40–60% GC content, optimum melting temperature (*Tm*) 68 °C, minimum *Tm* 60 °C and maximum difference in *Tm* between paired primers 2 °C. PAL_finder was set to search for sequences with a minimum of 8 repeat units ranging from di—to hexa—nucleotide repeats.

A total of 37 loci were selected for screening containing 8 tri-, 15 tetra-, 12 penta- and 2 hexa- nucleotides using 6 individuals to check for successful amplification and variation. Di-nucleotide repeats were not selected as allele scoring is generally more complicated for this repeat motif due to 'stutter bands' on either side of an allele peak (ascribed to enzyme slippage during amplification). This simulates allele peaks and therefore may lead to difficult and inaccurate scoring of alleles (*Guichoux et al., 2011*).

PCRs were carried out using the Type-it_Microsatellite PCR Kit (Qiagen, Venlo, Netherlands), with the recommended cycling conditions (5 min at 95 °C, 28× (30 s at 95 °C, 90 s at 60 °C, 30 s at 72 °C) and a final extension of 30 min at 60 °C). PCR products were initially analyzed using agarose gel electrophoresis, and loci were considered successful if one or two bands were present. Of the 37 initial loci, 10 successfully amplified according to these criteria. These loci were tested using labeled primers with florescent dyes VIC or 6-FAM in duplex PCRs (Table 1). A 3730 DNA Analyzer (Applied

Bertolotti et al. (2015), *PeerJ*, DOI 10.7717/peerj.1253

**Table 1  Characterization of ten polymorphic microsatellite loci for *Paralabrax albomaculatus*.**

| Locus | Genbank number | Primer sequence (5′–3′) | Repeat motif | Dye | Ta (°C) | Size range (bp) | Na | Ho | He |
|---|---|---|---|---|---|---|---|---|---|
| **PCR duplex set 1** | | | | | | | | | |
| Paxalb_10 | KP997010 | F: ACAAGTGCATCAAATACATGTCGG | ATCT (32) | 6-FAM | 63.8 | 404–480 | 24 | 0.919 | 0.944 |
| | | R: AAGGAATTCAATCTTAGTGGACACG | | | | | | | |
| Paxalb_4 | KP997008 | F: GCCTTATTCTCTCCTTTATCCCC | AAGAG (70) | VIC | 63.4 | 408–485 | 24 | 0.895 | 0.925 |
| | | R: CAAAGTTTTGAGACTGAGCAGGG | | | | | | | |
| **PCR duplex set 2** | | | | | | | | | |
| Paxalb_32 | KP997015 | F: ATGTCTTGCCTTATCTGTTGTGG | AAATT (45) | 6-FAM | 63.8 | 295–373 | 26 | 0.718 | 0.927 |
| | | R: ACTAAACAGCGACGTTATACGAGG | | | | | | | |
| Paxalb_22 | KP997013 | F: TCCCAACCAACACCATTTTATGGC | TTTC (56) | VIC | 66.2 | 305–454 | 21 | 0.914 | 0.922 |
| | | R: TCCCTCTCGTTCTCTCCGACTTGC | | | | | | | |
| **PCR duplex set 3** | | | | | | | | | |
| Paxalb_11 | KP997011 | F: GAGATGCTGGAGAACTCAGAGGGC | TGC (24) | 6-FAM | 68.2 | 189–259 | 19 | 0.964 | 0.871 |
| | | R: AACGACTCCGGCGATTCAGC | | | | | | | |
| Paxalb_1 | KP997007 | F :AACCATGATCACACCTCCATCTTCC | ATCT (88) | VIC | 67.4 | 305–445 | 44 | 0.935 | 0.966 |
| | | R: AGCCTTTATGTGGTGAAGGGGTGC | | | | | | | |
| **PCR duplex set 4** | | | | | | | | | |
| Paxalb_20 | KP997012 | F: CTGCATTGACAATCTATTGTTCTGG | AAAAC (75) | 6-FAM | 63.3 | 359–474 | 49 | 0.882 | 0.98 |
| | | R: GCACGGTGCAATATTTTCTTTCC | | | | | | | |
| Paxalb_24 | KP997014 | F: GTTTTGGTCCAGATGCTTTTAATGG | AAT (54) | VIC | 64 | 419–477 | 23 | 0.575 | 0.9 |
| | | R: ACTGTACTGGCTCCAACTGCTGC | | | | | | | |
| **PCR duplex set 5** | | | | | | | | | |
| Paxalb_8 | KP997009 | F: GATGTAGCCAGCACAGCAAATGACC | AAAG (68) | 6-FAM | 66.5 | 316–415 | 36 | 0.956 | 0.963 |
| | | R: CCTCCATCCTCAACTTTCTCAATTAAATCC | | | | | | | |
| Paxalb_35 | KP997016 | F: TGTTCCTCGCCTCAAAGTAGGACG | AAT (39) | VIC | 68.2 | 382–414 | 11 | 0.844 | 0.815 |
| | | R: CACCGATACAGACCTTTGACAGGC | | | | | | | |

**Notes.**

Duplex set, primers that were combined in one PCR; F, forward sequence; R, reverse sequence; Repeat motif, number of times the nucleotide motif is repeated; Dye, fluorescent dye used to label each primer; Ta, optimal annealing temperature; Na, number of alleles; Ho, observed heterozygosity; He, expected heterozygosity.

Biosystems, Carlsbad, California, USA) was used for the fragment length analysis of the PCR products with the Genescan$^{TM}$ 500 LIZ® size standard. Allele peaks were scored using GeneMapper® Software Version 3.7 (Applied Biosystems, Carlsbad, California, USA) following the procedure recommended by *Selkoe & Toonen (2006)*. Null alleles and scoring errors were checked using Microchecker version 2.2.3 (*Van Oosterhout et al., 2004*) and information regarding Hardy–Weinberg equilibrium (HWE) was tested using GenoDive (*Meirmans & Van Tienderen, 2004*). Finally, estimates of allele frequency for the set of microsatellites with null alleles were provided using FreeNA (*Chapuis & Estoup, 2007*).

## RESULTS

The 10 loci show high levels of polymorphism with 11–49 alleles per locus (Table 1). Microsatellites Pax_alb20, Pax_alb24 and Pax_ alb32 were characterized as containing possible null alleles, and deviated from HWE. Allele frequencies estimates for these 3 microsatellites were of 0.0987, 0.0401, and 0.1634, respectively.

## DISCUSSION

As no previous work has been carried out on this endemic species, these loci will be useful for further research to investigate population connectivity, structure and genetic diversity as well as help with the implementation of informed fisheries management.

Although three loci showed evidence for null alleles, estimates of null allele frequencies show that these loci are nonetheless useful for estimating genetic diversity. The high number of alleles per locus could mean that these 10 primers would be very useful to show variation between populations.

## ACKNOWLEDGEMENTS

This study was part of the Charles Darwin Station and the Galapagos National Park project "Ecology, evaluation and management of fisheries: steps towards sustainability" (POA PC 25-14). The authors would like to thank Jorge Baque and Isabel Haro for their valuable help in collecting samples as well as the fishermen of Santa Cruz Island for their collaboration. Special thanks to the Galapagos National Park and the Charles Darwin Research Station for providing us with permits and logistical support for this work.

### Funding

The University of Manchester, Smithsonian Marine Station and Charles Darwin Research Station provided funding for this work. The funders had no role in study design, data collection and analysis, decision to publish, or preparation of the manuscript.

### Grant Disclosures

The following grant information was disclosed by the authors:
University of Manchester.
Smithsonian Marine Station.
Charles Darwin Research Station.

## Competing Interests

Nathan K. Truelove and Stephen J. Box are employees of the Smithsonian Marine Station.

## Author Contributions

- Alicia C. Bertolotti conceived and designed the experiments, performed the experiments, analyzed the data, wrote the paper, prepared figures and/or tables, reviewed drafts of the paper, field data collection, experiment, analysis.
- Sarah M. Griffiths conceived and designed the experiments, performed the experiments, analyzed the data, reviewed drafts of the paper, experiment, analysis.
- Nathan K. Truelove conceived and designed the experiments, performed the experiments, analyzed the data, reviewed drafts of the paper, supervision.
- Stephen J. Box contributed reagents/materials/analysis tools, funding.
- Richard F. Preziosi conceived and designed the experiments, contributed reagents/materials/analysis tools, reviewed drafts of the paper, funding, supervision.
- Pelayo Salinas de Leon contributed reagents/materials/analysis tools, reviewed drafts of the paper, field work.

## Animal Ethics

The following information was supplied relating to ethical approvals (i.e., approving body and any reference numbers):

Galapagos National Park, Research permit number: PC-28-13. This permit allowed for the collection of samples for the species in this study within the Galapagos Marine Reserve.

## DNA Deposition

The following information was supplied regarding the deposition of DNA sequences:

GenBank. Accession numbers: KP997007– KP997016.

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
