# Peer review of "Isolation and characterization of 10 polymorphic microsatellite loci for the endangered Galapagos-endemic whitespotted sandbass (*Paralabrax albomaculatus)"

_PeerJ, doi:10.7717/peerj.1253_

## Round 0.1 · original submission · Minor Revisions

Please consider all the suggestions in the revised version of the manuscript.

·

Basic reporting

This paper describes the development of 10 microsatellite loci for a commercially important fish in the Galapagos Marine Reserve. Overall the methods appear appropriate and the microsatellites appear useful. However, several aspects of the text require improvement, as I describe below.

Please report the fragment length of sequenced reads (e.g. 150bp) and the number of sequence reads obtained in the raw data.

What kind of trimming was conducted in Trimmomatic? For example, were reads trimmed based on quality scores? If so, what were your quality score criteria? What were the numbers of sequence reads after trimming?

The authors state they used 70 samples from two geographic locations—how many samples were from each location? Instead of calling these geographic locations “distinct” (which is a bit vague), you might want to state the approximate geographic distance between the two locations.

For the loci exhibiting null alleles, it would be helpful for the authors to report estimates of the frequency of null alleles--e.g. this statistic can be calculated using the program FreeNA. This statistic would give readers some indication of the magnitude of the impact the null alleles might have on population genetics analyses.

Line 79: This sentence is awkward--should “Although” be changed to “Because”?
In the Discussion section, the authors imply that conducting a study using larger sample sizes will inform researchers as to whether the null alleles of 3 loci are problematic for population genetics analyses. It is unclear to me how increasing sample sizes would accomplish this. Please clarify.

I do not see any legend explaining what the column headings of the table mean—please add this.

Experimental design

No Comments

Validity of the findings

No comments

·

Basic reporting

The manuscript is clearly written and provides a very useful set of markers that will be helpful to the community. However, I consider that several minor revisions are needed before the manuscript can be accepted for publication as there is a general lack of details, some of which being critical to ensure repeatability of the experiments.

Experimental design

No comments

Validity of the findings

No comments

Additional comments

This manuscript can be published after minor revisions.
I have some specific comments through the text:
Line 46: as the sequencing length could restrain the available microsatellite motifs, it would be necessary to write which sequencing length was done (2x250bp v2 version or 2x300bp v3 version)
Line 56: no di-nucleotide repeat markers were kept for screening, the reader should know why the authors decided to restrict their selection (or maybe there is no di-nucleotide 8+ repeats for this species?)
Line 59: I could not find any information on the geographic structure of the samples, is it 35 from each population?
Line 61: what sequencer was used to run the electrophoresis and what size standard was used as calibration? What software was used to analyze the microsatellite profiles?
Line 61: I could not find anywhere which primer is labeled with which dye, neither in this part of the text nor in the table. The reader would need details about the duplexes so they are able to repeat the experiment themselves.
Line 64: I would move the sentence “Samples were collected…” to line 44 right before “To obtain microsatellite…” as it provides a required information at this point of the paper.
Line70: What method was used to test for Hardy-Weinberg equilibrium and with what software?
I am not familiar with microchecker but if its assumption is that null alleles are detected by HW departure then it sounds quite obvious that you would find the same 3 loci. In any case the authors should provide more details.
The table requires a detailed legend to provide standalone information where it is not self-explanatory (e.g. I suspect that ATCT (88) means that 88 is the number of repeats, but I may be wrong).
Finally the style used when listing the reference is not homogeneous and should be checked.

---

## Round 0.2 · accepted · Accept

Thank you for improving your manuscript.